# Research on the Impact of Farmland Transfer on Rural Household Consumption: Evidence from Yunnan Province, China

**Mingyong Hong and Lei Lou ***

School of Economics, Guizhou University, Guiyang 550025, China
* Correspondence: gs.llou21@gzu.edu.cn

**Abstract:** By constructing the analytical framework of "farmland transfer—farmland function—income structure—rural household consumption", based on the sample data of 537 rural households in 50 villages in Yunnan Province of China, this paper uses the OLS model to explore the impact of farmland transfer on rural household consumption and uses an intermediary effect model to further explore its internal transmission mechanism. The research finds that: (1) Farmland transfer (farmland transfer-out or farmland transfer-in) can stimulate rural household consumption. (2) The coefficient of farmland transfer-out to non-food consumption is 0.118, which is greater than its coefficient of food consumption of 0.016; the rural households of farmland transfer-out are more willing to increase non-food consumption expenditure, which is conducive to the optimization of their consumption structure. (3) The coefficient of farmland transfer-in to food consumption is 0.028, which is greater than its coefficient to non-food consumption of 0.009; the rural households of farmland transfer-in are more willing to increase food consumption expenditure, which is not conducive to the optimization of their consumption structure. (4) Rural household consumption expenditure will show a downward trend with the increase in the age of the head of the rural household, and the consumption structure will also show a deterioration. (5) The more family assets rural households have, the stronger their consumption expenditure capacity, which is conducive to optimizing their consumption structure. (6) The results of the intermediary effect model show that farmland transfer affects rural households' consumption and consumption structure by affecting rural households' income under different livelihood modes. Accordingly, the paper puts forward some suggestions on establishing the benefit coordination mechanism of farmland transfer, improving the non-agricultural employment mechanism of the rural surplus labor force, raising the expected return on farmland investment, increasing the proportion of household income saved appropriately and strengthening the social security mechanism in order to further promote the orderly transfer of farmland, improve the consumption capacity and consumption level of rural households, expand rural domestic demand and promote rural consumption upgrading.

**Keywords:** farmland transfer; farmland function; income structure; rural household consumption; consumption structure; Yunnan Province

## 1. Introduction

At present, the intensification of Sino-US trade contradictions has directly led to the increase of instability and uncertainty in China's foreign trade environment. In addition, the continuous impact of COVID-19 and the downward pressure of economic structural transformation have hindered the high-quality development of China's economy [1,2]. In response to this, on 10 April 2020, General Secretary Xi Jinping proposed at the seventh meeting of the Central Finance and Economics Commission to "build a new development pattern with a large domestic cycle as the mainstay and dual domestic and international cycles to promote each other" and take advantage of China's mega market and domestic demand potential. In December 2021, the Central Economic Work Conference stressed that

"we should deepen the structural reform on the supply side, focusing on unblocking the domestic circulation, breaking through the supply constraint blockage and opening up the links of production, distribution, circulation and consumption". It can be seen that there is still a huge potential space for consumption to drive China's economic development. However, studies by relevant scholars show that China's final consumption accounts for 54.3% of GDP in 2020, which is far below the global average share of 78.1% [3]. To achieve high-quality development, China's economy must seek to tap the potential of domestic demand, which is mainly derived from insufficient consumption [4,5], especially in the context of rural revitalization, the rural consumption market is promising [6]. To this end, the No. 1 document of the Central Government of China in 2021 emphasized that "we should comprehensively promote rural consumption, promote effective linkage between urban and rural production and consumption, and meet the needs of rural residents for consumption upgrading". Meanwhile, China's 14th Five-Year Plan specifies that in the next five years we should "improve the urban-rural integration consumption network, expand the coverage of e-commerce in rural areas, improve the consumption environment in counties and promote the upgrading of rural consumption ladder". However, data from the China National Bureau of Statistics show that the per capita consumption expenditure of rural residents in China in 2021 is 15,916 RMB yuan, while the consumption expenditure of urban residents in the same period is 30,307 RMB yuan, and the urban-rural expenditure ratio is 1.904[1], so the rural consumption market has endless potential [7,8]. Therefore, it is easy to see that the rural market will be the main town to tap the consumption space in China both now and in the future.

In July 2013, General Secretary Xi Jinping pointed out during his visit to the Wuhan Comprehensive Rural Property Rights Exchange that the ownership, contract right and management right of farmland (unless otherwise specified, farmland in this paper is equivalent to contracted land of rural households) should be separated. In November of the same year, the Third Plenary Session of the 18th CPC Central Committee resolved to establish a model of "separation of three rights" in China's agricultural management system [9–12], breaking the shackle that the management right of farmland could not be freely transferred. Since then, under the mandatory arrangement of a series of formal institutions, China's farmland transfer market has gradually developed and begun to take shape [13–16]. By the end of 2017, the area of contracted land of rural households transfer in China reached 512 million mu[2], accounting for 37% of the total area of family-operated arable land [17]. It is noteworthy that from 2013 to 2018, the per capita consumption expenditure of Chinese rural residents rose from 7485 RMB yuan to 12,124 RMB yuan, with an average annual growth rate of 10.12%. Coincidentally, in the year (2014) following the formal implementation of the "separation of three rights" model for farmland, the growth rate of per capita consumption expenditure of rural residents in China was as high as 12% (see Figure 1). Based on this, we can draw a general guess that there may be a certain correlation between the transfer of farmland and the consumption of rural households. In fact, relevant scholars have already paid attention to the possible impact of farmland transfer on rural household consumption. Based on the perspectives of farmland transfer-out, Xing and Chen [18], Chen et al. [19] and Shi and Zhu [20] pointed out that farmland transfer-out significantly increased the consumption level of rural households. A study by Yang et al. [21] based on the perspective of social capital showed that farmland transfer could influence the key natural capital changes and livelihood strategy adjustment of rural households, which positively and significantly promoted the consumption level of rural households, and rural households who participated in farmland transfer had higher consumption enthusiasm compared with those who did not engage in farmland transfer. Hu and Ding [22] used the regression analysis results of OLS and Quantile models with 7000 rural households in CFPS 2012, which showed that farmland transfer had heterogeneous effects on the consumption level of rural households with different characteristics, and only the complementary effects of farmland transfer and social security could effectively promote rural household consumption.

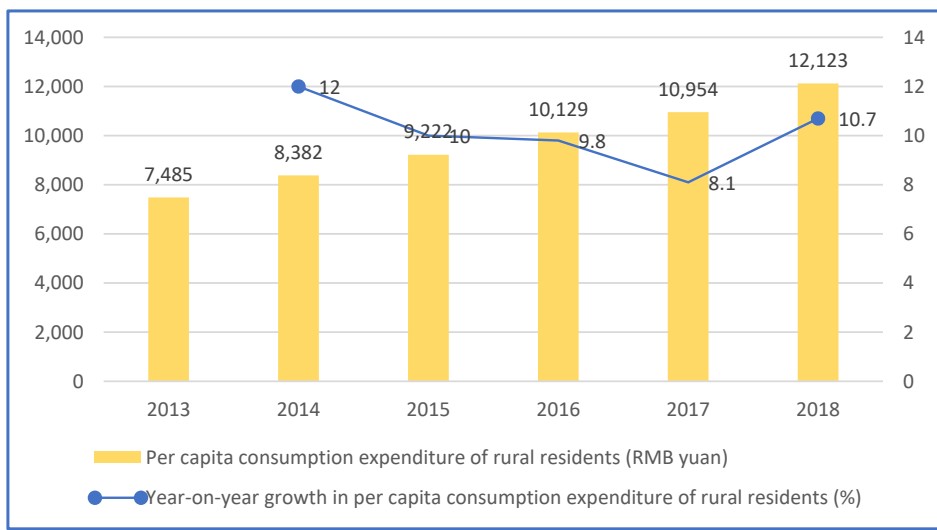

**Figure 1.** Trend of per capita consumption expenditure of rural residents in China[3].

Farmland transfer has received extensive academic attention because of the fundamental importance of agriculture, and a large number of studies have focused on the effects of farmland transfer on rural household income [23,24], rural poverty reduction [25,26], willingness to citizenship [27], food cultivation structure [28], willingness to livelihood transition [29] and have gradually transitioned to the effects on arable land quality protection [30,31], agricultural production efficiency [32], rural household entrepreneurial decisions [33] and other areas. There is a consensus in the academic community that enhancing the consumption capacity and consumption level of rural households is the finishing touch to expanding China's rural domestic demand [1]. Although scholars have conducted empirical studies on the impact of farmland transfer on rural household consumption, the relevant literature is still relatively scarce. In addition, in the relatively scarce papers, first, there is almost no systematic theoretical analysis framework to specifically elaborate the theoretical mechanism relationship between farmland transfer and rural household consumption; second, there is almost no use of a persuasive indicator like the Engel coefficient that can reflect the consumption structure of rural households to study the impact of farmland transfer on rural household consumption structure. In view of these, the impact of farmland transfer on rural households' consumption deserves further study. Therefore, the aims of the study are: First, we will construct a theoretical framework of "farmland transfer—farmland function—income structure—rural household consumption" to systematically explain the theoretical mechanism relationship between farmland transfer and rural household consumption. Second, by using first-hand research data of 537 rural households in 50 villages in Yunnan Province of China, we use the OLS model to explore the impact of farmland transfer on rural household consumption[4] and use the intermediary effect model to further explore its internal transmission mechanism. Third, because the Engel coefficient can reflect the characteristics of consumption structure, so we dichotomize the total consumption expenditure of rural households into two types of expenditure, food consumption and non-food consumption to further investigate how farmland transfer affects the consumption structure of rural households.

## 2. Theoretical Analysis and Research Hypotheses

Theoretically, the income level of rural households and the level of social security entitlement are the two fundamental factors that influence rural households' consumption [1]. Farmland, as the largest livelihood, influences the general social structural characteristics of most rural households in China [21]. Since the reform and opening up, with the rapid progress of urbanization, industrialization and agricultural modernization, farmers have broken free from the shackles of farmland to gradually enter the cities and towns for non-

agricultural employment. The small-scale, loose and fragmented farmland management model is no longer able to meet the needs of rural economic development [34]. For this reason, the Chinese government has been making efforts in top-level design, formulating and issuing a series of relevant policy documents to strengthen rural households' residual claims to farmland, relax the management rights of farmland and guarantee the realization of farmland transfer benefits for rural households. Undoubtedly, the transfer of farmland can promote the optimal allocation of land resources and moderate scale operation of agriculture, allow the rational and adequate allocation of rural factors of production such as land, labor, technology and capital, effectively promote the development of the rural economy and the improvement of the income level of rural households and completely activate the productive and property functions of farmland [9–12]. However, it must be acknowledged that the social security system in China's rural areas is not yet sound, and the transfer of farmland can indeed change the fate of the rural households concerned to a certain extent. In addition, farmland is increasingly becoming a basic survival guarantee for vulnerable groups of rural households who lack the ability to move to urban areas [35–37]. Of course, the security function of farmland is the most important and basic function of farmland for rural households. Whether it is the "further" property function of farmland or the "step back" productive function of farmland, when rural households realize the functional differentiation of farmland, not only does the security function of farmland not become lost [1] but also it increasingly strengthens the property and productive functions of farmland [38,39].

At present, the transfer of farmland is gradually becoming a new way for rural households to accumulate original capital. Based on the perspective of rural household livelihood of farmland transfer-out, the income effect of farmland property function can be divided into direct effect and indirect effect [40]. Among them, the direct effect is the rental income brought by the lease or transfer of farmland management rights to rural households of farmland transfer-out, and the income from the direct effect is agricultural income. However, at this time, the dependence of rural households on the agricultural income brought by farmland transfer-out is weak. While the indirect effect refers to the wage income obtained by farmland transfer-out to promote the transfer of surplus rural labor to the non-agricultural sector for employment. The rental income of farmland is an important part of transfer income, which has the characteristics of temporary income and rural households will be more casual in spending [3]. Wage income has a permanent character, and rural households prefer to use this income as a recurrent consumption expenditure [1]. It is obvious that the income structure of rural households is enriched and diversified by the property function of farmland. In addition, the theory of "psychological accounts" suggests that rural households can allocate different incomes to different accounts, which cannot be filled by each other, and that rural households have different consumption tendencies for different sources of income [41]. The enrichment of rural households' income structure is essentially the division of their holistic income into numerous units, which will greatly strengthen the perception of subjective wealth increase [1]. Therefore, the change in income structure brought about by the transfer-out of farmland can stimulate the consumption of rural households with both rental income and wage income [42]. At the same time, as a component reflecting the hierarchy of rural households' needs and the order of their satisfaction, food consumption is a demand dominated by rural households' physiological requirements, while non-food consumption is a pursuit of rural households' convenience and performance needs and personal enjoyment and development needs [43]. As the transfer-out of farmland gives rural households a richer income structure and brings them a higher level of subjective income, they will gradually reduce food consumption to satisfy their physiological needs and increase non-food consumption of goods and services for convenience and performance needs as well as personal enjoyment and development needs [44]. Thus, the transfer-out of farmland can lead to an increase in the non-food consumption capacity of rural households, which in turn helps to optimize the consumption structure of rural households.

The perspective of the rural household livelihood strategy is based on farmland transfer-in, rural households mainly focus on agricultural production, and farmland has become one of the most important means of production for them. At this time, the dependence of rural households on agricultural income is very strong, and they will do everything possible to expand the scale of farmland to increase agricultural productive income. Therefore, farmland transfer promotes the rational and optimal allocation of land resources, allowing ordinary rural households to acquire relatively concentrated farmland, which is beneficial to a certain extent to the development of the agricultural production of rural households of farmland transfer-in in the direction of moderate scale, intensification, specialization or marketization and rural households realize income growth in the realization of productive functions of farmland and continuously improve their consumption capacity [40]. However, along with the basic completion of China's farmland titling and certification work, on the one hand, farmland titling has properly solved the problems of inaccurate area of contracted land of rural households parcels and unclear four directions, and the "public domain" of farmland property rights has been priced into the market, and rural households of farmland transfer-in have lost the organizational space to earn the "public domain" of property rights [9–12]. On the other hand, the stable property rights of farmland encourage the impersonalization and high rent of farmland transfer among acquaintances [45], which enhances the bargaining position and bargaining power of farmland transfer-out transactions of rural households. As a result, farmland transfer-in of rural households based on the productive use of farmland is faced with the dilemma of increasing production expenditure due to the expansion of production scale, while the income from single-structure agricultural production increases. The theory of "loss aversion" suggests that rural households feel more strongly when weighing losses than gains [1]. In addition, farming is a tough occupation, and rural households value hard-earned income [3]. In view of this, rural households are reluctant to increase their non-food consumption of goods and services for convenience and performance needs as well as personal enjoyment and development needs [44], resulting in a slowdown or even a decrease in the growth rate of non-food consumption expenditure [46]. Instead, rural households tend to increase their spending on food consumption to satisfy physiological needs [44], which is not conducive to optimizing the consumption structure of rural households.

Based on the above theoretical analysis, this paper constructs an analytical framework for the impact of farmland transfer on rural households' consumption (see Figure 2). Meanwhile, the following research hypothesis 1, hypothesis 2 and hypothesis 3 are proposed.

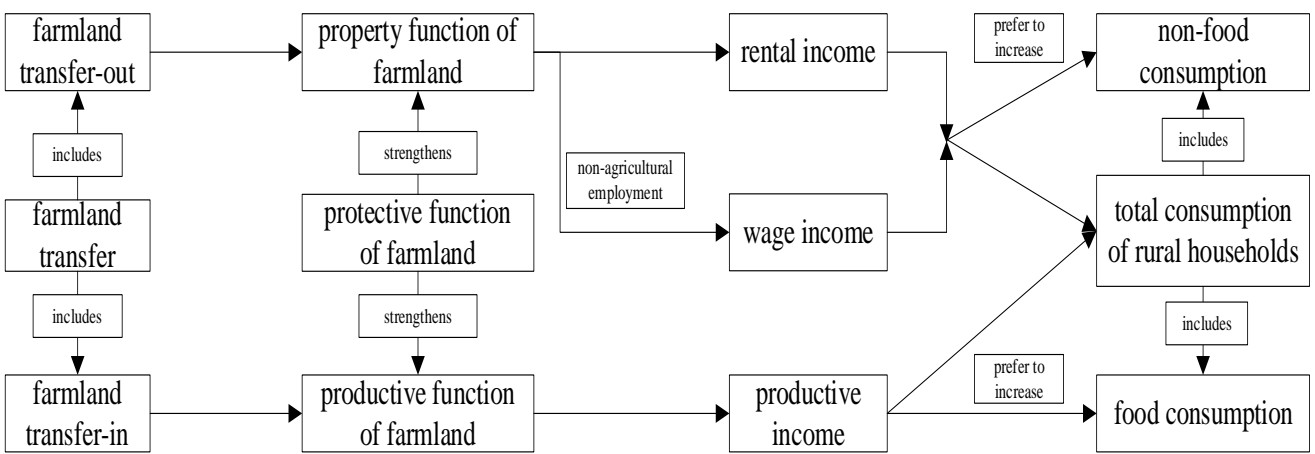

**Figure 2.** Analytical framework of the impact of farmland transfer on rural household consumption.

**H1:** *The transfer-out of farmland from rural households stimulates their consumption and makes them more willing to increase their non-food consumption expenditures, and then contributes to the optimization of their consumption structure.*

**H2:** *The transfer-in of farmland from rural households stimulates their consumption and makes them more willing to increase their food consumption expenditure, and then does not contribute to the optimization of their consumption structure.*

**H3:** *The transfer of farmland (farmland transfer-out or farmland transfer-in) from rural households affects their income under different livelihoods, and then affects their consumption and consumption structure.*

## 3. Research Design

### 3.1. Description of Selected Research Site and Data Sources

3.1.1. Description of Selected Research Site

Yunnan Province, located in southwest China, is an important part of the Yunnan-Kweichow Plateau and a relatively underdeveloped area in China. Compared with Guizhou Province, where 62% of the land area is karst landform, Yunnan Province has its comparative advantage in landform, which makes the farmland in Yunnan Province more valuable for circulation. In addition, although Yunnan Province is not the main grain-producing area in China, it is of great practical significance to study the impact of farmland transfer behavior on rural household consumption in plateau mountainous and underdeveloped areas. Based on the above explanations, we selected Yunnan Province as the final research point of this paper.

3.1.2. Data Sources

To explore the impact of farmland transfer on rural households' consumption, in November 2021, teachers, doctoral students and master's students of related majors from the School of Economics of Guizhou University and the School of Economics of Yunnan University, formed a relevant subject group to conduct rural household surveys in 16 prefecture-level cities or autonomous prefectures in Yunnan Province. In order to reduce sampling bias, the research team used a stratified random sampling method, stratified according to the administrative vertical relationship of the city (state)—county (city, district)—township (town)—village in turn. One county (city or district) was randomly selected in each prefecture-level city or autonomous prefecture, two townships or towns were randomly selected in each county (city or district), one to two villages were randomly selected in each township or town, and 10 to 15 questionnaires were randomly distributed to rural households in each village under investigation. In addition, this paper takes 2020 as a unit time cycle and a key time node of this survey, so as to facilitate the interview of relevant issues and data collection and collation by the members of the research group. Finally, in this survey, 650 questionnaires were distributed in 50 villages, and 600 questionnaires were recovered with a recovery rate of 92.31%. In addition, out of the 600 questionnaires collected, the questionnaires with obvious errors, repeated relevant content and inconsistent with the research theme of this paper were discarded. Finally, 537 valid questionnaires were obtained, involving 50 villages, with an effective rate of 82.62%.

### 3.2. Variable Settings

3.2.1. Explained Variables

The China National Bureau of Statistics categorizes the consumption of rural residents in China into eight major types, including food, clothing, housing, household equipment and supplies, transportation and communication, education and entertainment, health care and other consumption. Besides this, rural household and family are both organizational concepts. Unless otherwise specified, the number of rural households and family members in this paper is consistent. We use the 2020 per capita household consumption expenditure (logarithmicized) to represent the total consumption expenditure of rural households

according to China's national statistical caliber and drawing on the research practices of Geng et al. [1], Chen et al. [19] and Yang et al. [21]. However, unlike Cai et al. [47] who covered per capita household consumption expenditure divided into three types of expenditure: subsistence consumption, developmental consumption and productive consumption to measure the consumption expenditure and consumption structure of rural households, this paper uses this feature of the Engel coefficient to reflect changes in consumption structure to dichotomize per capita household consumption expenditure in 2020 into two types of expenditure: per capita rural household food consumption and per capita rural household non-food consumption. As we all know, the Engel coefficient refers to the proportion of food consumption expenditure in the total consumption expenditure of the family. It was put forward by Engel, a German statistician in the 19th century, on the change of consumption structure based on empirical statistical data. Generally speaking, the smaller the Engel coefficient, the better the consumption structure of the family. In other words, the more the family spends on non-food consumption, the better the consumption structure of the family will be.

### 3.2.2. Explanatory Variables

The modes of farmland transfer are more complex, including lease, exchange, transfer, equity and etc. Drawing on the research results of Chen et al. [19] and Yang et al. [21], this paper uses farmland transfer-out and farmland transfer-in to measure farmland transfer: (1) Farmland transfer-out, the question in the research questionnaire is "In 2020, did your family transfer-out contracted land to others?". The relevant values are: 1 = yes and 0 = no. (2) Farmland transfer-in, means "In 2020, did your family transfer in contracted land from other people or collectives, excluding your own contracted land?". If the answer is "yes", the value is 1, and if not, the value is 0.

### 3.2.3. Mediating Variable

In order to test whether farmland transfer (farmland transfer-out or farmland transfer-in) affects the income of rural households under different livelihoods, and then affects the consumption structure of rural households. In the questionnaire, we set the question "What is the annual income of your family in 2020 by choosing the corresponding livelihood mode through the transfer of farmland (farmland transfer-out or farmland transfer-in)?" to identify this. In addition, the logarithm of per capita household income in 2020 was used to indicate the income of rural households under different livelihood options.

### 3.2.4. Control Variables

To mitigate the omission of variables that lead to biased estimation results, this paper also includes variables at the level of household head characteristics [1], family characteristics [19] and village characteristics [21] that affect rural household consumption as control variables. Among them, household head characteristics include the gender of the household head, age of the household head and marriage of the household head; family characteristics include the number of family members, age per capita of the family and assets per capita of the family (logarithmicized); and village characteristics include whether there is non-agricultural economy in the village, the availability of public transportation in the village, the topographical condition of the village, and the distance from the village to the county. In addition, the unit of consumption and asset-related variables is RMB yuan, and there is no unit in the value assignment of variables after logarithmic processing. The specific relevant variable settings and statistical descriptions are shown in Table 1.

**Table 1.** Variable definitions and descriptive statistics.

| Dimension | Variable Name | Variable Assignment | Mean | Standard Deviation |
|---|---|---|---|---|
| Explained variables | Total consumption of rural households | ln (1 + per capita household consumption expenditure) | 9.487 | 0.746 |
| | Food consumption | 1n (1 + per capita household consumption expenditure on food) | 8.445 | 0.807 |
| | Non-food consumption | 1n (1 + household per capita non-food consumption expenditure) | 8.871 | 1.008 |
| Explanatory variables | Farmland transfer-out | 1 = yes, 0 = no | 0.340 | 0.474 |
| | Farmland transfer-in | 1 = yes, 0 = no | 0.375 | 0.485 |
| Mediating variable | rural household income | ln (1 + household income per capita) | 9.809 | 1.381 |
| Control variables | Gender of household head | 1 = male, 0 = female | 0.677 | 0.480 |
| | Age of household head | age | 49.907 | 9.969 |
| | Marriage of household head | 1 = married, 0 = unmarried | 0.981 | 0.172 |
| | Number of family members | person | 3.752 | 1.378 |
| | Family age per capita | age | 39.802 | 12.000 |
| | Family assets per capita | ln (1 + family assets per capita) | 11.356 | 1.042 |
| | Whether there is non-agricultural economy in the village | 1 = with non-agricultural economy, 0 = without non-agricultural economy | 0.601 | 0.494 |
| | Availability of public transportation in the village | 1 = with public transportation, 0 = without public transportation | 0.662 | 0.473 |
| | Topographical conditions of the village | 1 = flat land, 2 = sloping land | 1.483 | 0.511 |
| | Distance from the village to the county | km | 27.645 | 20.256 |

*3.3. Model Selection*

Because this paper mainly explores the impact of farmland transfer on rural household consumption, it is appropriate to use an OLS model for estimation. To this end, a relevant benchmark model is constructed by drawing on the research practices of Dong and Huang [48] and Hu and Ding [22], which has the following basic form:

$$CS_i = \beta_0 + \beta_1 X_i + \beta_2 D_i + \varepsilon_{i1} \tag{1}$$

In Equation (1), $CS_i$ denotes the total consumption of rural households, food consumption and non-food consumption. $X_i$ denotes farmland transfer-out and farmland transfer-in. $D_i$ denotes a matrix of control variables, including household head characteristics, family characteristics and village characteristics. $\beta_0$ is a constant term, $\beta_1$ and $\beta_2$ are coefficients to be estimated and $\varepsilon_{i1}$ denotes an error term and is assumed to satisfy a standard normal distribution.

To test whether farmland transfer (farmland transfer-out or farmland transfer-in) acts on rural households' consumption and consumption structure through the path of influencing rural households' income under different livelihoods. Then, this paper draws on the study of Wen and Ye [49] and further constructs an intermediary effect model based on model (1) with rural households' income under different livelihoods as the mediating variable as follows:

$$FI_i = \delta_0 + \delta_1 X_i + \delta_2 D_i + \varepsilon_{i2} \tag{2}$$

$$CS_i = \phi_0 + \phi_1 X_i + \phi_2 FI_i + \phi_3 D_i + \varepsilon_{i3} \tag{3}$$

In the above model, $FI_i$ is the mediating variable, representing rural households' income under different livelihoods; $\delta_0$ and $\phi_0$ are constant terms, $\delta_1$, $\delta_2$, $\phi_1$, $\phi_2$ and $\phi_3$ are coefficients to be estimated, $\varepsilon_{i2}$ and $\varepsilon_{i3}$ denote error terms and are assumed to satisfy standard normal distribution; other variables and coefficients are defined in the same way as Equation (1).

## 4. Results and Analysis

### 4.1. Multicollinearity Test

Since the introduction of more variables at the level of household head characteristics, family characteristics and village characteristics in this paper may pose the problem of multicollinearity, variance inflation factor (VIF) is used for multicollinearity diagnosis. In Table 2, the results show that the variance inflation factor (VIF) values are all less than 10, and we can judge that there is no more serious multicollinearity problem basically.

**Table 2.** Multicollinearity test.

| Variable Name | Total Consumption of Rural Households | | Food Consumption | | Non-Food Consumption | |
|---|---|---|---|---|---|---|
| | VIF | VIF | VIF | VIF | VIF | VIF |
| Farmland transfer-out | 1.401 | | 1.412 | | 1.461 | |
| Farmland transfer-in | | 1.343 | | 1.329 | | 1.394 |
| Rural household income | 1.261 | 1.216 | 1.217 | 1.319 | 1.378 | 1.301 |
| Gender of household head | 1.231 | 1.097 | 1.283 | 1.271 | 1.269 | 1.265 |
| Age of household head | 1.116 | 1.112 | 1.328 | 1.374 | 1.325 | 1.451 |
| Marriage of household head | 1.383 | 1.262 | 1.391 | 1.471 | 1.271 | 1.308 |
| Number of family members | 1.365 | 1.341 | 1.413 | 1.296 | 1.523 | 1.357 |
| Family age per capita | 1.219 | 1.258 | 1.258 | 1.365 | 1.579 | 1.426 |
| Family assets per capita | 1.187 | 1.096 | 1.143 | 1.429 | 1.421 | 1.329 |
| Whether there is non-agricultural economy in the village | 1.236 | 1.163 | 1.274 | 1.385 | 1.438 | 1.075 |
| Availability of public transportation in the village | 1.291 | 1.061 | 1.381 | 1.279 | 1.219 | 1.091 |
| Topographical conditions of the village | 1.363 | 1.247 | 1.227 | 1.394 | 1.105 | 1.208 |
| Distance from the village to the county | 1.348 | 1.119 | 1.421 | 1.283 | 1.194 | 1.364 |

### 4.2. Benchmark Regression Results

#### 4.2.1. Analysis of the Impact of Farmland Transfer-Out on Rural Household Consumption

In Table 3, the coefficients of the farmland transfer-out variable are significantly positive, and the coefficient of non-food consumption is 0.118, which is larger than the coefficient of food consumption is 0.016, which verifies hypothesis 1 of this paper, that is, farmland transfer-out can stimulate the consumption of rural households, and rural households are more willing to increase their non-food consumption expenditure, which is beneficial to the optimization of rural consumption structure. The "psychological accounts" theory states that people categorize their income into different accounts according to the way they receive it, which are mutually exclusive and not complementary, and that different income patterns result in different consumption tendencies [1]. The income structure will become richer as rural households generally receive farmland rental income and wage income after their farmland is transferred out, which will continuously strengthen the subjective wealth effect of rural households and induce them to consume. In addition, after rural households transfer out of farmland, they will generally move away from the countryside to engage in non-agricultural production activities in the city. Affected by the new consumption habits of the surrounding people, rural households who transfer-out farmland will gradually change their original consumption habits that prefer to increase food consumption expenditure to those that are more willing to increase non-food consumption expenditure. Therefore, when rural households satisfy the surplus of food consumption expenditure, they are more willing to increase the expenditure on non-food consumption, and their consumption structure will be optimized accordingly.

**Table 3.** Impact of farmland transfer on rural household consumption: Results of benchmark regression.

| Variable Name | Total Consumption of Rural Households | | Food Consumption | | Non-Food Consumption | |
|---|---|---|---|---|---|---|
| Farmland transfer-out | 0.083 ** | | 0.016 ** | | 0.118 ** | |
| | (2.454) | | (2.323) | | (2.367) | |
| Farmland transfer-in | | 0.017 ** | | 0.028 ** | | 0.009 ** |
| | | (2.445) | | (2.312) | | (2.327) |
| Gender of household head | 0.020 | −0.065 | −0.072 | 0.001 | −0.010 | −0.021 |
| | (0.237) | (−0.961) | (−1.068) | (0.015) | (−0.167) | (−0.362) |
| Age of household head | −0.023 *** | −0.021 *** | −0.010 *** | −0.016 *** | −0.017 *** | −0.018 *** |
| | (−5.467) | (−3.040) | (−2.899) | (−5.166) | (−5.616) | (−5.367) |
| Marriage of household head | 0.107 | −0.129 | −0.136 | 0.114 | 0.029 | 0.032 |
| | (0.466) | (−0.691) | (−0.730) | (0.499) | (0.183) | (0.202) |
| Number of family members | −0.020 | −0.063 | −0.063 | −0.024 | −0.030 | −0.033 |
| | (−0.665) | (−0.154) | (−0.156) | (−0.801) | (−1.430) | (−1.537) |
| Family age per capita | −0.010 | −0.007 | −0.008 | −0.004 | −0.092 | −0.004 |
| | (−0.149) | (−1.364) | (−1.234) | (−1.071) | (−1.241) | (−0.086) |
| Family assets per capita | 0.298 *** | 0.209 *** | 0.107 *** | 0.184 *** | 0.189 *** | 0.192 *** |
| | (4.574) | (5.921) | (5.897) | (4.394) | (6.213) | (6.064) |
| Whether there is non-agricultural economy in the village | −0.041 | −0.004 | −0.005 | −0.057 | −0.034 | −0.043 |
| | (−0.493) | (−0.061) | (−0.067) | (−0.694) | (−0.583) | (−0.740) |
| Availability of public transportation in the village | −0.099 | −0.047 | −0.047 | −0.081 | −0.054 | −0.045 |
| | (−1.136) | (−0.660) | (−0.665) | (−0.936) | (−0.894) | (−0.736) |
| Topographical conditions of the village | −0.015 | 0.036 | 0.032 | −0.013 | −0.001 | −0.001 |
| | (−0.174) | (0.525) | (0.452) | (−0.155) | (−0.015) | (−0.009) |
| Distance from the village to the county | 0.002 | 0.000 | 0.000 | 0.000 | 0.000 | −0.001 |
| | (0.913) | (0.200) | (0.204) | (0.218) | (0.188) | (−0.342) |
| -Cons | 7.634 *** | 8.821 *** | 8.443 *** | 7.387 *** | 11.343 *** | 7.936 *** |
| | (8.638) | (10.561) | (10.560) | (8.691) | (14.313) | (14.394) |
| N | 537 | 537 | 537 | 537 | 537 | 537 |

Note: *** and ** refer to the statistics being significant at the 1% and 5% levels, respectively. Inside the regression parentheses are *t* values of coefficients.

### 4.2.2. Analysis of the Impact of Farmland Transfer-In on Rural Household Consumption

The coefficients of farmland transfer-in variables are significantly positive, and the coefficient of food consumption is 0.028, which is larger than the coefficient of non-food consumption is 0.009. Hypothesis 2 of this paper that farmland transfer can stimulate rural household consumption but rural households who transfer-in farmland are more willing to increase their expenditure on food consumption, which is not beneficial to the optimization of their consumption structure, is confirmed. Schultz's rational theory of small farmers shows that small farmers are poor and efficient, that is, farmers are people with entrepreneurial spirit and can use the right resources [22]. Rural households who transfer-in farmland may engage in moderate scale operation, take advantage of the scale of farmland to reduce the cost of agricultural production, give full play to the rational and effective allocation of resources such as labor and agricultural machinery for agricultural production to bring about an increase in production efficiency, improve the income of agricultural production of rural households and enhance the consumption capacity of rural households. As a matter of fact, agriculture is a very difficult occupation, and rural households cherish the income that is difficult to obtain. In addition, rural households tend to have a high propensity to save preventively for a single productive income from agriculture [47]. After rural households transfer-in farmland, they are still mainly engaged in agricultural production. The consumption habits of the surrounding people and themselves will not change much. Rural households who transfer-in farmland will still maintain their original consumption habits and are more willing to increase food consumption expenditure than

non-food expenditure. Therefore, the increase in income obtained from the transfer-in of farmland to rural households will increase their consumption capacity to a certain extent, but they will save after satisfying the surplus of food consumption expenditure and are generally unwilling to spend too much on non-food consumption, which makes it difficult to optimize their consumption structure.

Furthermore, based on the communication with rural households in the field survey, we know that rural households are mainly engaged in agricultural production before the transfer of farmland. Although the total consumption expenditure of rural households will increase, rural households are more inclined to increase food consumption expenditure, which is not conducive to the optimization of rural households' consumption structure. After the transfer of farmland, the total consumption expenditure of rural households will continue to increase, but rural households who transfer-in farmland are more willing to increase food consumption expenditure, which is not conducive to the optimization of their consumption structure. The rural households that transfer-out farmland are more willing to increase non-food consumption expenditure, which is beneficial to the optimization of their consumption structure. Therefore, the farmland transfer has a heterogeneous impact on the consumption expenditure and consumption structure of rural households of the farmland transfer-out and rural households of the farmland transfer-in.

### 4.2.3. Analysis of the Impact of Control Variables on Rural Household Consumption

The coefficients of the household head's age variable are all significantly negative, and their coefficients on food consumption are larger than those on non-food consumption, that is, the consumption expenditure of rural households tends to decline as the household head gets older, and their consumption structure also shows a deterioration. The possible explanation is that in rural Chinese society, the head of the household is the mainstay of the family and his income is the most important source of income for the rural household. The coefficients of the family assets per capita variable are all significantly positive, and their coefficients on non-food consumption are larger than those on food consumption, indicating that the more family assets, rural households have the stronger consumption capacity and the more conducive to optimizing their consumption structure. The possible reason for this is that family assets have a certain "wealth effect" and "asset effect", which can bring a stable income stream to rural households, thus enhancing their consumption ability and improving their consumption structure [48].

### *4.3. Robustness Test and Endogeneity Discussion*

### 4.3.1. Robustness Test I: Sub-Sample Test

In the field research, we found that a few rural households have two-way farmland transfer behaviors of both farmland transfer-out and farmland transfer-in. However, mixing rural households' two-way farmland transfer behaviors with one-way farmland transfer behavior for regression estimation may affect the authenticity of the results. For this reason, drawing on the study of Yang et al. [21], the data of a sample of 16 rural households with both farmland transfer-out and farmland transfer-in 2020 are excluded from the subsample test. The results in Table 4 show that the significance level of coefficients and the sign of coefficients of the farmland transfer-out and farmland transfer-in variables and the magnitude of coefficients between them on food consumption and on non-food consumption variables are consistent with the results of the benchmark regression, indicating that the benchmark regression results are robust.

**Table 4.** Robustness test I: Sub-sample test.

| Variable Name | Total Consumption of Rural Households | | Food Consumption | | Non-Food Consumption | |
|---|---|---|---|---|---|---|
| Farmland transfer-out | 0.081 ** | | 0.018 ** | | 0.119 ** | |
| | (2.426) | | (2.318) | | (2.347) | |
| Farmland transfer-in | | 0.016 ** | | 0.029 ** | | 0.008 ** |
| | | (2.432) | | (2.351) | | (2.358) |
| Control variables | Yes | Yes | Yes | Yes | Yes | Yes |
| -Cons | 7.595 *** | 8.812 *** | 8.424 *** | 7.408 *** | 11.348 *** | 7.913 *** |
| | (8.701) | (10.596) | (10.243) | (8.638) | (14.254) | (14.313) |
| N | 521 | 521 | 521 | 521 | 521 | 521 |

Note: *** and ** refer to the statistics being significant at the 1% and 5% levels, respectively. Inside of regression parentheses are *t* values of coefficients. Control variables are kept consistent with Table 3.

### 4.3.2. Robustness Test II: Replacing Core Explanatory Variables and Re-Estimating

To exclude the estimation bias caused by measurement bias, this paper uses the method of Hu and Ding [22] to conduct robustness tests using the average per mu income from farmland transfer-out and the average per mu expenditure from farmland transfer-in as proxies for farmland transfer-out and farmland transfer-in[5], respectively. The results in Table 5 show that the significance levels of the coefficients and the sign of the coefficients of the variables of the average per mu income from farmland transfer-out and the average per mu expenditure from farmland transfer-in and the magnitudes of the coefficients between the variables of food consumption and non-food consumption are consistent with the results of the benchmark regression, indicating that the results of the benchmark regression are robust.

**Table 5.** Robustness test II: Replacement of core explanatory variables.

| Variable Name | Total Consumption of Rural Households | | Food Consumption | | Non-Food Consumption | |
|---|---|---|---|---|---|---|
| The average per mu income from farmland transfer-out | 0.016 ** | | 0.009 ** | | 0.019 ** | |
| | ((2.362)) | | ((2.543)) | | ((2.436)) | |
| The average per mu expenditure from farmland transfer-in | | 0.007 ** | | 0.009 ** | | 0.005 ** |
| | | (2.392) | | (2.385) | | (2.521) |
| Control variables | Yes | Yes | Yes | Yes | Yes | Yes |
| -Cons | 6.527 *** | 7.697 *** | 7.493 *** | 6.467 *** | 10.396 *** | 6.921 *** |
| | (8.576) | (9.989) | (10.542) | (9.634) | (11.357) | (11.186) |
| N | 537 | 537 | 537 | 537 | 537 | 537 |

Note: *** and ** refer to the statistics being significant at the 1% and 5% levels, respectively. Inside of regression parentheses are *t* values of coefficients. Control variables are kept consistent with Table 3.

### 4.3.3. Robustness Test III: Re-Estimation Using Propensity Matching Score Method

To eliminate the endogeneity problem caused by the possible selectivity bias of the sample, this paper uses the propensity matching score method (PSM) for robustness testing [50]. Based on Table 3 control variables matching control and experimental groups, rural households of farmland transfer-out and farmland transfer-in are set as the experimental group, and rural households of farmland non-transfer-out and farmland non-transfer-in are set as the control group. The average treatment effects (ATT) of farmland transfer-out and farmland transfer-in are estimated using nearest neighbor matching, radius matching and kernel matching, respectively. The results of the common support condition test of Figure 3 show that most of the observations are within the common range of values when matching using the three matching methods of nearest neighbor matching (k = 4), radius matching (caliper = 4) and kernel matching (bwidth = 0.06), and thus the matching quality is reliable.

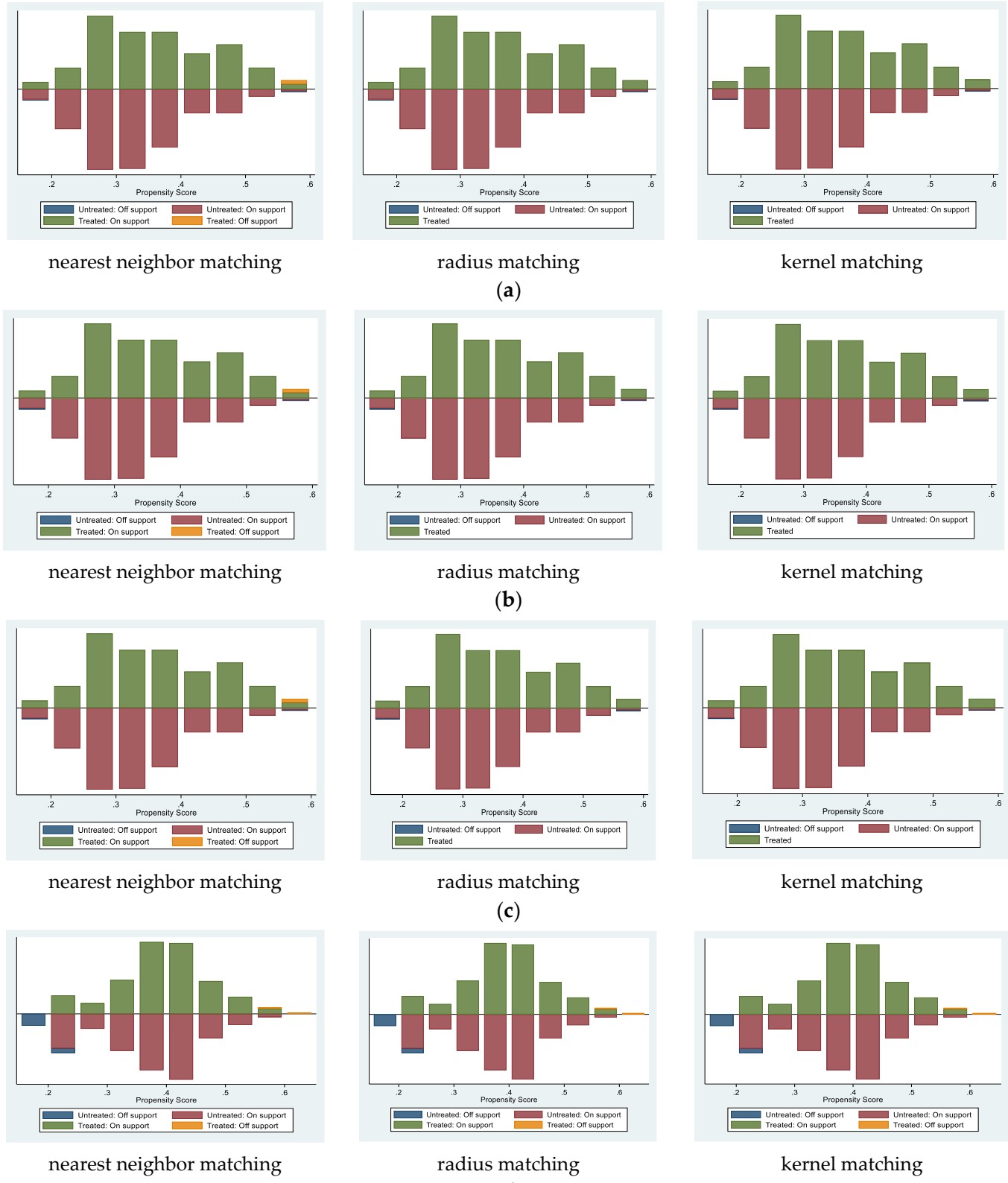

**Figure 3.** *Cont.*

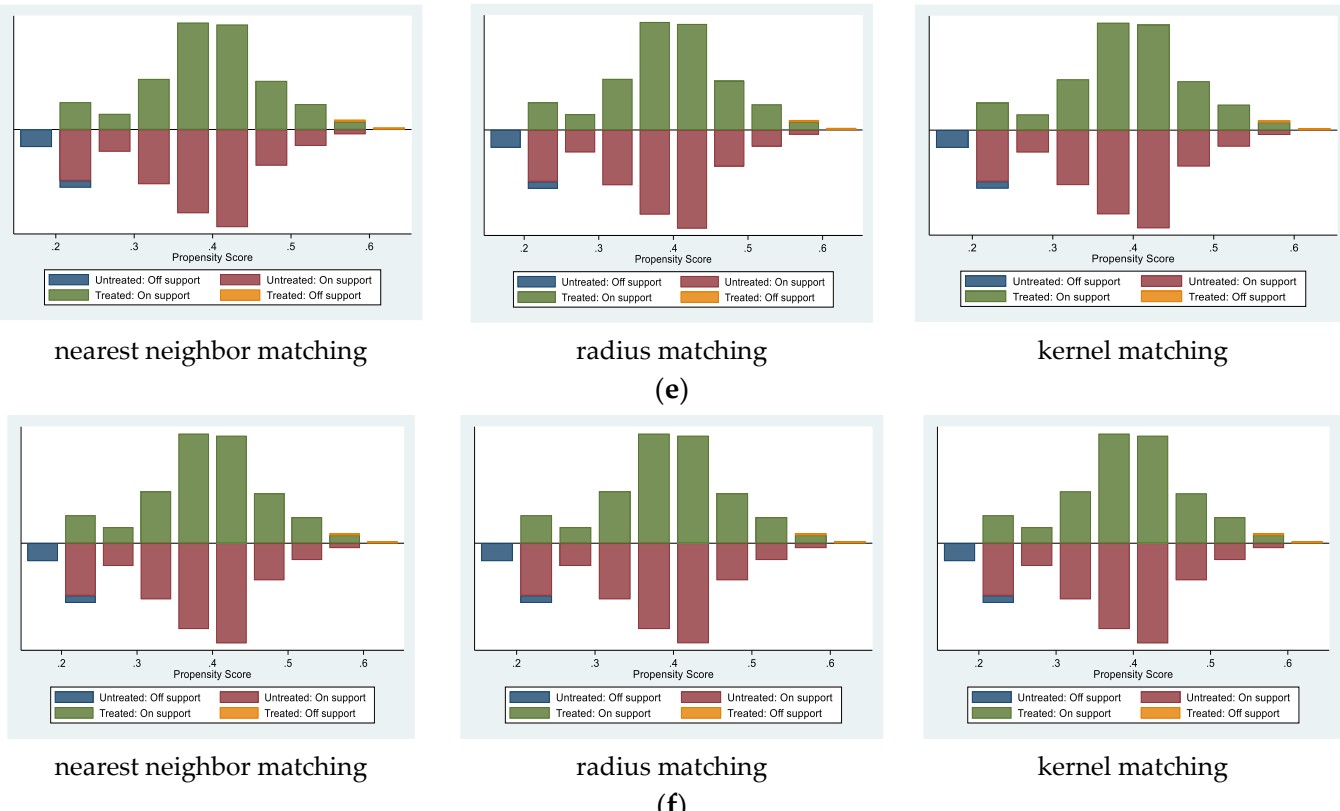

**Figure 3.** Propensity score distribution and the common support for propensity score estimation. (**a**) farmland transfer-out and farmland not transfer-out, *CS* = total consumption of rural households. (**b**) farmland transfer-out and farmland not transfer-out, *CS* = food consumption. (**c**) farmland transfer-out and farmland not transfer-out, *CS* = non-food consumption. (**d**) farmland transfer-in and farmland not transfer-in, *CS* = total consumption of rural households. (**e**) farmland transfer-in and farmland not transfer-in, *CS* = food consumption. (**f**) farmland transfer-in and farmland not transfer-in, *CS* = non-food consumption.

The results in Table 6 show that the average treatment effects obtained by the nearest neighbor matching, radius matching and kernel matching methods provide further evidence that either farmland transfer-out or farmland transfer-in significantly enhances the consumption capacity of rural households. In addition, taking the nearest neighbor matching method as an example, after excluding other factors, the per capita non-food consumption expenditure of rural households transferred out farmland will increase by 4.081% (exp (0.040) − 1), which is larger than the per capita food consumption expenditure by 0.602% (exp (0.006) − 1), and the per capita food consumption expenditure of rural households transferred in farmland will increase by 1.207% (exp (0.012) − 1), which is larger than the per capita non-food consumption expenditure by 0.401% (exp (0.004) − 1). Therefore, the re-estimation results based on the propensity matching score method (PSM) show that the benchmark regression results are robust.

**Table 6.** Robustness test III: Re-estimation using PSM.

| Variable Name | Matching Methods | ATT(Farmland Transfer-out) | *t*-Value | ATT(Farmland Transfer-In) | *t*-Value |
|---|---|---|---|---|---|
| Total consumption of rural households | nearest neighbor matching | 0.030 *** | 3.224 | 0.007 *** | 3.278 |
| | radius matching | 0.029 *** | 3.486 | 0.006 *** | 3.316 |
| | kernel matching | 0.029 *** | 3.218 | 0.006 *** | 3.265 |
| Food consumption | nearest neighbor matching | 0.006 *** | 3.212 | 0.012 *** | 3.223 |
| | radius matching | 0.006 *** | 3.317 | 0.011 *** | 3.468 |
| | kernel matching | 0.005 *** | 3.236 | 0.010 *** | 3.384 |
| Non-food consumption | nearest neighbor matching | 0.040 *** | 3.238 | 0.004 *** | 3.341 |
| | radius matching | 0.039 *** | 3.311 | 0.004 *** | 3.408 |
| | kernel matching | 0.038 *** | 3.289 | 0.003 *** | 3.227 |

Note: *** refers to the statistics being significant at the 1% level. Control variables are kept consistent with Table 3.

### 4.3.4. Robustness Test IV: Re-Estimation Using Instrumental Variable Method

When examining the impact of farmland transfer on rural household consumption, there may be endogeneity problems caused by reverse causality and omitted variables, and then the direct use of the OLS estimation method is likely to cause bias in model estimation. For this reason, this paper attempts to construct an instrumental variable model to eliminate the endogeneity problem caused by reverse causality and omitted variables. Drawing on the research results of Yang et al. [21] and Hu and Ding [22], the two-stage least squares (2SLS) estimation is conducted using "village farmland transfer-out rate" and "village farmland transfer-in rate" as the instrumental variables for farmland transfer-out and farmland transfer-in. As we all know, a qualified instrumental variable must satisfy two conditions, namely, the instrumental variable is highly correlated with the endogenous explanatory variables (correlation) and the instrumental variable is uncorrelated with the disturbance term (exogeneity). In this paper, the "village farmland transfer-out rate" and "village farmland transfer-in rate" are calculated based on the level of farmland transfer-out and the level of farmland transfer-in in the surveyed villages, which satisfy the requirements of correlation and exogeneity of the instrumental variable. The results in Table 7 show that the one-stage F values are all much greater than 10, indicating that the model does not have the problem of weak instrumental variables. The DWH values reject the original hypothesis that farmland transfer-out and farmland transfer-in are exogenous variables at the 1% level, indicating that the model has endogeneity problems. However, after correcting for the endogeneity problem induced by reverse causality and omitted variables, the significance level of coefficients and the sign of coefficients of farmland transfer-out and farmland transfer-in variables and the magnitude of coefficients between them on food consumption and on non-food consumption variables are consistent with the results of the benchmark regression, which verifies the credibility of the benchmark regression results.

**Table 7.** Robustness test IV: Re-estimation using instrumental variables method.

| Variable Name | Total Consumption of Rural Households | | Food Consumption | | Non-Food Consumption | |
|---|---|---|---|---|---|---|
| Farmland transfer-out | 0.064 ** | | 0.011 ** | | 0.116 ** | |
| | (2.468) | | (2.357) | | (2.349) | |
| Farmland transfer-in | | 0.016 ** | | 0.023 ** | | 0.007 ** |
| | | (2.412) | | (2.316) | | (2.363) |
| Control variables | Yes | Yes | Yes | Yes | Yes | Yes |
| Adj R$^2$ | 0.341 | 0.295 | 0.234 | 0.213 | 0.263 | 0.245 |
| The one-stage F-value | 100.781 | 99.867 | 96.483 | 96.538 | 95.892 | 97.346 |
| DWH-Chi$^2$ | 10.212 *** | 10.028 *** | 9.863 *** | 9.816 *** | 9.647 *** | 9.829 *** |
| N | 537 | 537 | 537 | 537 | 537 | 537 |

Note: *** and ** refer to the statistics being significant at the 1% and 5% levels, respectively. Inside the regression parentheses are *t* values of coefficients. Control variables are kept consistent with Table 3.

*4.4. Endogeneity Discussion*

The endogeneity problem is mainly caused by measurement error, selectivity bias, omitted variables and reverse causality [51]. For the measurement error problem, this paper solves it by replacing the core explanatory variables. For the selectivity bias problem, this paper mitigates it by using the propensity matching score method (PSM), which enables the observations to effectively avoid the estimation bias caused by sample self-selection through matching and resampling [52], thus improving the accuracy of the estimation results. For the omitted variables and reverse causality problems, this paper eliminates them by using the instrumental variables method, while adding as many control variables as possible to exclude the influence of omitted observables on the estimation results of this paper. In addition, the sub-sample of 16 households with both farmland transfer-out and farmland transfer-in is excluded for re-estimation to exclude the influence of different samples with different sensitivity to the obtained results. In summary, strictly speaking, there is no particularly serious endogeneity problem in this paper.

*4.5. Mechanism of Action: Intermediary Effect Test*

The results of the benchmark regressions and robustness tests indicate that farmland transfer (farmland transfer-out or farmland transfer-in) can stimulate rural household consumption, but there is heterogeneity in its effect on the consumption structure of rural households of farmland transfer-out and rural households of farmland transfer-in. Here, we further use the intermediary effect model to test its internal transmission mechanism. However, whether farmland is transferred out or transferred in actually represents the choice of livelihood modes of different rural households. As a result, the income of rural households who transfer-out farmland mainly includes rental income and wage income, and the income of rural households who transfer-in farmland mainly includes productive income. Therefore, this paper is not to test the intermediary effect of rent income, wage income and productive income on rural household consumption in the transfer of farmland, but to test the intermediary effect of income from the farmland transfer-out of rural households (the sum of rent income and wage income) on rural households' consumption and productive income from the farmland transfer-in of rural households on rural households' consumption. The results in Table 8 show that there is a significant positive effect of farmland transfer on the income of rural households under different livelihoods, indicating that rural households after farmland transfer can bring in stable income based on different livelihood strategies. In addition, the fitted regression results show that rural household income under different livelihoods positively affects total consumption of rural households, food consumption and non-food consumption at the 1% level of significance, which indicates that the intermediary effect of rural household income under different livelihoods exists and is significant. Besides this, the optimized consumption structure of rural households of farmland transfer-out and the deteriorated consumption structure of rural households of farmland transfer-in remain consistent with the benchmark regression results. That is, the impact path of "farmland transfer—rural households' income under different livelihoods—rural household consumption" holds. Through calculation, it is found that the intermediary effects of farmland transfer-out on the total consumption of rural households, food consumption and non-food consumption by affecting the income of rural households of farmland transfer-out are 39.014% (39.014% is obtained by multiplying the coefficient 0.257 of the farmland transfer-out variable to the rural household income variable in Table 8 by the coefficient 0.126 of the direct effect of the rural household income variable to the total consumption of rural households variable, and then dividing it by the coefficient 0.083 of the farmland transfer-out variable to the total consumption of rural households variable in Table 3, and then multiplying it by 100%. The rest of the intermediary effect proportion data can be obtained according to this calculation method), 30.519% and 44.648%, respectively; the intermediary effects of farmland transfer-in on the total consumption of rural households, food consumption and non-food consumption by affecting the income of rural households of farmland transfer-in are 38.912%, 40.250%

and 35.389%, respectively, which show that the intermediary effect accounts for a large proportion in the total utility of affecting the total consumption of rural households, food consumption and non-food consumption. The income of rural households based on different livelihood modes is a transmission mechanism that can not be ignored in the impact of farmland transfer on the rural household consumption, respectively. In addition, under different livelihood strategies, rural households' dependence on agricultural income is different, which may also be an important potential reason for the optimization of rural households who transfer-out farmland consumption structure and the deterioration of rural households who transfer-in farmland consumption structure. The intermediary effect of this paper is a partial intermediary effect.

**Table 8.** Intermediary effect test.

| Variable Name | Rural Household Income | | Total Consumption of Rural Households | | Food Consumption | | Non-Food Consumption | |
|---|---|---|---|---|---|---|---|---|
| Farmland transfer-out | 0.257 *** | | 0.051 ** | | 0.011 ** | | 0.065 ** | |
| | (3.518) | | (2.337) | | (2.362) | | (2.464) | |
| Farmland transfer-in | | 0.245 *** | | 0.010 ** | | 0.017 ** | | 0.006 ** |
| | | (3.459) | | (2.351) | | (2.348) | | (2.336) |
| Rural household income | | | 0.126 *** | 0.027 *** | 0.019 *** | 0.046 *** | 0.205 *** | 0.013 *** |
| | | | (3.351) | (3.421) | (3.462) | (3.475) | (3.373) | (3.648) |
| Control variables | Yes | Yes | Yes | Yes | Yes | Yes | Yes | Yes |
| -Cons | 7.621 *** | 8.232 *** | 6.476 *** | 8.342 *** | 8.411 *** | 7.253 *** | 9.243 *** | 7.814 *** |
| | (6.232) | (6.838) | (8.325) | (9.187) | (9.904) | (8.362) | (13.473) | (13.857) |
| N | 537 | 537 | 537 | 537 | 537 | 537 | 537 | 537 |

Note: *** and ** refer to the statistics being significant at the 1% and 5% levels, respectively. Inside the regression parentheses are *t* values of coefficients. Control variables are kept consistent with Table 3.

At the same time, a non-parametric percentile bootstrap sampling method with bias correction is used to conduct 5000 sampling tests to examine the intermediary effect of rural households' income under different livelihoods. In Table 9, the value of $\delta_1 \times \phi_2$ does not contain 0 at a 95% confidence interval, and the coefficients of $\delta_1$, $\phi_1$ and $\phi_2$ pass the 5% significance level test, and $\delta_1 \times \phi_2$ has the same sign as $\phi_1$, which indicate that the income of rural households under different livelihoods plays a part in the intermediary effect of farmland transfer on rural households' consumption and consumption structure, thus the results of the intermediary effect model test in this paper are valid and robust.

**Table 9.** Robustness test of intermediary effect.

| Coefficient | Farmland Transfer-Out | | | Farmland Transfer-In | | |
|---|---|---|---|---|---|---|
| | Total Consumption of Rural Households | Food Consumption | Non-Food Consumption | Total Consumption of Rural Households | Food Consumption | Non-Food Consumption |
| $\beta_1$ | 0.083 ** (2.454) | 0.016 ** (2.323) | 0.118 ** (2.367) | 0.017 ** (2.445) | 0.028 ** (2.312) | 0.009 ** (2.327) |
| $\delta_1$ | | 0.248 *** (3.186) | | | 0.235 *** (3.672) | |
| $\phi_2$ | 0.137 *** (3.867) | 0.021 *** (3.652) | 0.211 *** (3.034) | 0.031 *** (3.651) | 0.049 *** (3.439) | 0.017 *** (3.758) |
| $\delta_1 \times \phi_2$ | 0.034 | 0.005 | 0.052 | 0.007 | 0.012 | 0.004 |
| $\delta_1 \times \phi_2$ (95% Boot CI) | 0.0013~0.0126 | 0.0002~0.026 | 0.0113~0.2212 | 0.0021~0.0301 | 0.0036~0.0512 | 0.0016~0.0213 |
| $\Phi_1$ | 0.049 ** (2.353) | 0.011 ** (2.325) | 0.066 ** (2.375) | 0.010 ** (2.363) | 0.016 ** (2.298) | 0.005 ** (2.362) |
| Control variables | Yes | Yes | Yes | Yes | Yes | Yes |
| -Cons | 8.634 *** (7.654) | 8.975 *** (8.134) | 7.908 *** (9.079) | 8.768 *** (9.908) | 9.031 *** (12.136) | 8.902 *** (11.784) |
| Test conclusion | Partial intermediary effect | Partial intermediary effect | Partial intermediary effect | Partial intermediary effect | Partial intermediary effect | Partial intermediary effect |

Note: *** and ** refer to the statistics being significant at the 1% and 5% levels, respectively. Inside the regression parentheses are *t* values of coefficients. Control variables are kept consistent with Table 3.

## 5. Discussion

In this section, we discuss the potential contributions and limitations of this research.

The first discussion concerns the major contributions to the existing literature. This paper contributes to the current studies in four ways. (1) We use the survey data of 537 rural households in 50 villages in Yunnan Province, which is relatively underdeveloped in Southwest China and is located in the Yunnan-Kweichow Plateau, mainly in plateau and mountain terrain, to study the impact of farmland transfer on rural household consumption, which has unique regional characteristics and greater practical significance. (2) By constructing an analytical framework of "farmland transfer—farmland function—income structure—rural household consumption", we comprehensively analyzed the theoretical mechanism relationship between farmland transfer and rural household consumption. (3) Although there is a small amount of literature on the impact of farmland transfer on rural household consumption, this paper more systematically studies the impact of farmland transfer on rural household consumption through benchmark regression, robustness test and intermediary effect test. At the same time, we have achieved more fruitful study results. (4) Based on the empirical study of 537 rural households in 50 villages in Yunnan Province, we have obtained some new findings. For example, rural household consumption expenditure will show a downward trend with the increase in the age of the head of rural household, and the consumption structure will also show a deterioration. Another example is that the more family assets, rural households have the stronger consumption expenditure capacity, which is conducive to optimizing their consumption structure.

The second discussion is about the limitations of this study. (1) The results of this study are based on the corresponding empirical analysis of 537 rural households survey data in 50 villages in Yunnan Province. There are certain regional limitations, and whether it is applicable to other regions remains to be discussed, but the significance of the results of this study is not to be underestimated. (2) Based on the cross-sectional data of 537 households in 50 villages in Yunnan Province in 2020, the research conclusion is that the static impact of farmland transfer on rural household consumption and consumption structure cannot reflect the trend of time dynamic impact of farmland transfer on rural household consumption and consumption structure. This requires our team to conduct a continuous follow-up survey on these rural households and use panel data to overcome the limitation of this study.

## 6. Conclusions and Policy Implications

### 6.1. Conclusions

Based on the first-hand survey data of 537 rural households in 50 villages in Yunnan Province, this paper constructs an analytical framework of "farmland transfer—farmland function—income structure—rural household consumption", uses the OLS model to deeply explore the impact of farmland transfer on rural household consumption, and further uses the intermediary effect model to explore its internal transmission mechanism. The following conclusions are drawn: First, farmland transfer (farmland transfer-out or farmland transfer-in) can stimulate rural household consumption. Second, the coefficient of farmland transfer-out to non-food consumption is 0.118, which is larger than the coefficient of farmland transfer-out to food consumption is 0.016; rural households who transfer-out farmland are more willing to increase non-food consumption expenditure, which is beneficial to the optimization of their consumption structure. Third, the coefficient of farmland transfer-in to food consumption is 0.028, which is larger than its coefficient of non-food consumption is 0.009; rural households who transfer-in farmland are more willing to increase food consumption expenditure, which is not conducive to the optimization of their consumption structure. The above research results are still robust after excluding possible endogenous problems through four robustness tests, namely, sub-sample test, replacement core explanatory variables test, propensity matching score (PSM) test and instrumental variable test, which shows that the conclusions obtained from benchmark regression are true and reliable to a large extent. Fourth, rural household consumption expenditure will show a downward

trend with the increase in the age of the head of the rural household, and the consumption structure will also show a deterioration. Fifth, the more family assets, rural households have the stronger consumption expenditure capacity, which is conducive to optimizing their consumption structure. Sixth, the results of the intermediary effect model show that the transfer of farmland affects rural households' consumption and consumption structure by affecting rural households' income under different livelihood modes. At the same time, using the non-parametric percentile bootstrap sampling method of deviation correction, the results of 5000 sampling tests show that the effect of the intermediary effect model is effective and robust.

### 6.2. Policy Implications

Improving the consumption capacity and consumption level of rural households is not only a strong response to the major strategic deployment of "accelerating the construction of a new development pattern with domestic big cycle as the main body and domestic and international double cycles promoting each other" put forward in China's 14th Five-Year Plan, but also conducive to the orderly advancement of China National New Urbanization Plan and China Rural Revitalization Strategy. Therefore, in order to further release the consumption capacity of rural households and improve their consumption level, this paper draws the following enlightenment: first, it is necessary to establish the interest coordination mechanism of farmland transfer, constantly reduce the transaction cost of farmland transfer and guide rural households to carry out farmland transfer in an orderly manner, so as to realize the optimal allocation of farmland resources. Second, improve the non-agricultural employment mechanism of rural surplus labor force, reasonably arrange rural households of farmland transfer-out and strengthen their skills training, so as to ensure the stability of their non-agricultural employment and obtain higher income. Third, improve the stability of farmland property rights, promote rural households of farmland transfer-in for moderate scale operation and constantly encourage them to improve the expected return on investment in farmland, so as to ensure the sustained and stable growth of their agricultural production. Fourth, social security shoulders the major responsibility of ensuring people's livelihood, promoting social equity and meeting the needs of the people for a better life. In the new era, rural areas should build a multi-level social security system in an all-around way, so as to lay a foundation for promoting the improvement of rural households' consumption ability and the optimization of consumption structure. Fifth, rural households should save an appropriate amount of their income and appropriately increase their family assets.

**Author Contributions:** Econometric analysis and writing the original draft, L.L.; conceptual formulation, survey design, data collection, modeling, manuscript editing and revision, M.H. and L.L. All authors have read and agreed to the published version of the manuscript.

**Funding:** This research was funded by the National Natural Science Foundation of China (72163003), Three Dimensional Practice Model for Cultivating Innovative and Entrepreneurial Talents in Agriculture and Forestry (2020346) and the Foundation of Postgraduate of Guizhou Province (YJSKYJJ[2021]035).

**Institutional Review Board Statement:** Not applicable.

**Informed Consent Statement:** Not applicable.

**Data Availability Statement:** Not applicable.

**Conflicts of Interest:** The authors declare no conflict of interest.

## Notes

1    Data source: http://www.stats.gov.cn/tjsj/sjjd/202201/t20220118_1826529.html (accessed on 12 October 2022).
2    1 mu = 1/15 hectare.
3    Data source: «China Statistical Yearbook» (2014–2019).
4    In this paper, rural household consumption of rural households, food consumption and non-food consumption; among them, total consumption expenditure of rural households is the sum of food consumption expenditure and non-food consumption expenditure.
5    ln (1 + the average per mu income from farmland transfer-out) and ln (1 + the average per mu expenditure from farmland transfer-in) are used to define the average per mu income from farmland transfer-out and the average per mu expenditure from farmland transfer-in, respectively.

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
