# Peer review of "Research on the Impact of Farmland Transfer on Rural Household Consumption: Evidence from Yunnan Province, China"

_land, doi:10.3390/land11122147_

Round 1

Reviewer 1 Report (Previous Reviewer 2)

It can be seen that the authors worked on the article. They tried to meet all the reviewer's requirements. They added new charts and pictures. In this form, the article is more attractive to the reader. I think the article can be supported for publication in Land magazine. I recommend publishing.

Author Response

About the modification description of the article "Research on the Impact of Farmland Transfer on Rural Household Consumption: Evidence from Yunnan Province, China"

The first peer review expert doesn't put forward some suggestions, as follows:

It can be seen that the authors worked on the article. They tried to meet all the reviewer's requirements. They added new charts and pictures. In this form, the article is more attractive to the reader. I think the article can be supported for publication in Land magazine. I recommend publishing.

Thank you very much for the recognition of our revision by the first peer review expert. It is your recognition and encouragement that make us work harder on the academic road.

Kind regards,

Mingyong Hong, Lei Lou.

Reviewer 2 Report (New Reviewer)

Please add the following:

1. What is the aim/objective of the study?

2. What is the main novelty/contribution?

3. Better explain data and variables used.

4. What can be the study limitations and why?

Author Response

About the modification description of the article "Research on the Impact of Farmland Transfer on Rural Household Consumption: Evidence from Yunnan Province, China"

We would like to thank the second peer review expert for the constructive comments. On the basis of carefully understanding the opinions of the second peer review expert, we revised the article (the revised part is marked in blue or red in the revised version). Next, we will make some explanations to the second peer review experts about the revision.

The second peer review expert put forward four constructive suggestions, as follows:

The first:

Please add the following: What is the aim/objective of the study?

Modification description: We have made the supplementary notes in the part of Introduction, as follows:

Therefore, the aims of the study are: First, we will construct a theoretical framework of “farmland transfer - farmland function - income structure - rural household consumption” to systematically explain the theoretical mechanism relationship between farmland transfer and rural household consumption. Second, by using first-hand research data of 537 rural households in 50 villages in Yunnan Province of China, we use OLS model to explore the impact of farmland transfer on rural household consumption3, and use intermediary effect model to further explore its internal transmission mechanism. Third, because Engel coefficient can reflect the characteristics of consumption structure, so we dichotomize the total consumption expenditure of rural households into two types of expenditure, food consumption and non-food consumption to further investigate how farmland transfer affects the consumption structure of rural households.

The second:

Please add the following: What is the main novelty/contribution?

Modification description: We have made the supplementary notes in the part of Disscusion , as follows:

This paper contributes to the current studies in four ways. (1) We use the survey data of 537 rural households in 50 villages in Yunnan Province, which is relatively underdeveloped in Southwest China and are located in Yunnan-Kweichow Plateau, mainly in plateau and mountain terrain, to study the impact of farmland transfer on rural household consumption, which has unique regional characteristics and greater practical significance. (2) By constructing an analytical framework of “farmland transfer - farmland function - income structure - rural household consumption”, we comprehensively analyzed the theoretical mechanism relationship between farmland transfer and rural household consumption. (3) Although there is a small amount of literature on the impact of farmland transfer on rural household consumption, this paper more systematically studies the impact of farmland transfer on rural household consumption through benchmark regression, robustness test and intermediary effect test. At the same time, we have achieved more fruitful study results. (4) Based on the empirical study of 537 rural households in 50 villages in Yunnan Province, we have obtained some new findings. For example, rural household consumption expenditure will show a downward trend with the increase of the age of the head of rural household, and the consumption structure will also show a deterioration. Another example is that the more family assets, rural households have the stronger consumption expenditure capacity, which is conducive to optimizing their consumption structure. 

The Third:

Please add the following: Better explain data and variables used.

Modification description: We have made the supplementary notes in the part of Research Design to better explain data and variables used, as follows:

  • in this survey, 650 questionnaires were distributed in 50 villages, and 600 questionnaires were recovered with a recovery rate of 92.31%. In addition, out of the 600 questionnaires collected, the questionnaires with obvious errors, repeated relevant content and inconsistent with the research theme of this paper were discarded. Finally, 537 valid questionnaires were obtained, involving 50 villages, with an effective rate of 82.62%.
  • Besides, rural household and family are both organizational concepts. Unless otherwise specified, the number of rural household and family members in this paper is consistent.
  • this paper uses farmland transfer-out and farmland transfer-in to measure farmland transfer.
  • In the questionnaire, we set the question of “what is the annual income of your family in 2020 by choosing the corresponding livelihood mode through the transfer of farmland (farmland transfer-out or farmland transfer-in)” to identify.
  • In adition, the unit of consumption and asset related variables is RMB yuan, and there is no unit in the value assignment of variables after logarithmic processing.

The fourth:

Please add the following: What can be the study limitations and why?

Modification description: We have made the supplementary notes in the part of Disscusion, as follows:

The second discussion is about the limitations of this study. (1) The results of this study are based on the corresponding empirical analysis of 537 rural households survey data in 50 villages in Yunnan Province. There are certain regional limitations, and whether it is applicable to other regions remains to be discussed, but the significance of the results of this study is not to be underestimated. (2) Based on the cross-sectional data of 537 households in 50 villages in Yunnan Province in 2020, the research conclusion is that the static impact of farmland transfer on rural household consumption and consumption structure cannot reflect the trend of time dynamic impact of farmland transfer on rural household consumption and consumption structure. This requires our team to conduct a continuous follow-up survey on these rural households and use panel data to overcome the limitation of this study.

Finally, We would like to thank the second peer review expert for the constructive comments. We hope that our revised article can be published in the journal of Land very much.

Kind regards,

Mingyong Hong, Lei Lou.

Reviewer 3 Report (New Reviewer)

1 The article uses one-year data of farmland transfer to compare the consumption level and consumption structure of farming households. To better evaluate the impact of farmland transfer on farm household consumption, it is suggested to consider the influence of farming households' consumption habits on consumption structure and add a longitudinal comparison of farming households' consumption changes before and after farmland transfer.

2 The conclusions of this article do not fully summarize all the conclusions of the OLS results. I suggest you enrich the conclusions according to your analysis above.   

3 Compared with the OLS results analysis, the tests of robustness are too long. It is recommended to add the analysis of your basis regression and discuss more the results of OLS.

4 Some similar articles have discussed fully farmland transfer and household consumption with similar methods. So, what is new or innovative? It is advised to discuss the coefficient differences between the regions involved in your research according to income or consumption level. It may be a way to acquire some different conclusions in the research.

Author Response

About the modification description of the article "Research on the Impact of Farmland Transfer on Rural Household Consumption: Evidence from Yunnan Province, China"

We would like to thank the third peer review expert for the constructive comments. On the basis of carefully understanding the opinions of the third peer review expert, we revised the article (the revised part is marked in blue or red in the revised version). Next, we will make some explanations to the third peer review experts about the revision.

The third peer review expert put forward four constructive suggestions, as follows: 

The first:

The article uses one-year data of farmland transfer to compare the consumption level and consumption structure of farming households. To better evaluate the impact of farmland transfer on farm household consumption, it is suggested to consider the influence of farming households' consumption habits on consumption structure and add a longitudinal comparison of farming households' consumption changes before and after farmland transfer.

Modification description: In order to consider the impact of rural households' consumption habits on consumption structure, we add the following contents in the part of Benchmark Regression Results.

In addition, after rural households transfer out farmland, they will generally move away from the countryside to engage in non-agricultural production activities in the city. Affected by the new consumption habits of the surrounding people, rural households who transfer out farmland will gradually change their original consumption habits that prefer to increase food consumption expenditure to those that are more willing increase non-food consumption expenditure.

After rural households transfer in farmland, they are still mainly engaged in agricultural production. The consumption habits of the surrounding people and themselves will not change much. Rural households who transfer in farmland will still maintain their original consumption habits and are more willing to increase food consumption expenditure than non-food expenditure.

We also add a longitudinal comparison of farming households' consumption changes before and after farmland transfer in the part of Benchmark Regression Results, as follows:

Furthermore, based on the communication with rural households in the field survey, we know that rural households are mainly engaged in agricultural production before the transfer of farmland. Although the total consumption expenditure of rural households will increase, rural households are more inclined to increase food consumption expenditure, which is not conducive to the optimization of rural households' consumption structure. After the transfer of farmland, the total consumption expenditure of rural households will continue to increase, but rural households who transfer in farmland are more willing to increase food consumption expenditure, which is not conducive to the optimization of their consumption structure; the rural households who transfer out farmland are more willing to increase non-food consumption expenditure, which is beneficial to the optimization of their consumption structure; therefore, the farmland transfer has a heterogeneous impact on the consumption expenditure and consumption structure of rural households of the farmland transfer-out and rural households of the farmland transfer-in.

The second:

The conclusions of this article do not fully summarize all the conclusions of the OLS results. I suggest you enrich the conclusions according to your analysis above.

Modification description: We have enriched our research conclusions in the part of Conclusions, as follows:

Fourth, rural household consumption expenditure will show a downward trend with the increase of the age of the head of rural household, and the consumption structure will also show a deterioration. Fifth, the more family assets, rural households have the stronger consumption expenditure capacity, which is conducive to optimizing their consumption structure.

At the same time, based on the newly added conclusions, we also enrich our policy implications in the part of Policy Implications

Fourth, social security shoulders the major responsibility of ensuring people’s livelihood, promoting social equity and meeting the needs of the people for a better life. In the new era, rural areas should build a multi-level social security system in an all-round way, so as to lay a foundation for promoting the improvement of rural households’ consumption ability and the optimization of consumption structure. Fifth, rural households should save an appropriate amount of their income and appropriately increase their family assets.

The third:

Compared with the OLS results analysis, the tests of robustness are too long. It is recommended to add the analysis of your basis regression and discuss more the results of OLS.

Modification description: We have added the analysis in the part of Benchmark Regression Results, as follows:

  • In addition, after rural householdstransfer out farmland, they will generally move away from the countryside to engage in non-agricultural production activities in the city. Affected by the new consumption habits of the surrounding people, rural households who transfer out farmland will gradually change their original consumption habits that prefer to increase food consumption expenditure to those that are more willing increase non-food consumption expenditure.
  • After rural householdstransfer in farmland, they are still mainly engaged in agricultural production. The consumption habits of the surrounding people and themselves will not change much. Rural households who transfer in farmland will still maintain their original consumption habits and are more willing to increase food consumption expenditure than non-food expenditure.
  • Furthermore, based on the communication with rural households in the field survey, we know that rural householdsare mainly engaged in agricultural production before the transfer of farmland. Although the total consumption expenditure of rural households will increase, rural households are more inclined to increase food consumption expenditure, which is not conducive to the optimization of rural households' consumption structure. After the transfer of farmland, the total consumption expenditure of rural households will continue to increase, but rural households who transfer in farmland are more willing to increase food consumption expenditure, which is not conducive to the optimization of their consumption structure; the rural households who transfer out farmland are more willing to increase non-food consumption expenditure, which is beneficial to the optimization of their consumption structure; therefore, the farmland transfer has a heterogeneous impact on the consumption expenditure and consumption structure of rural households of the farmland transfer-out and rural households of the farmland transfer-in.
  • The possible explanation is that in the rural Chinese society, the head of the household is the mainstay of the family and his income is the most important source of income for the rural household.
  • indicating that the more family assets, rural householdshave the stronger consumption capacity and the more conducive to optimizing their consumption structure.

The fourth:

Some similar articles have discussed fully farmland transfer and household consumption with similar methods. So, what is new or innovative? It is advised to discuss the coefficient differences between the regions involved in your research according to income or consumption level. It may be a way to acquire some different conclusions in the research.

Modification description: According to the third peer review’s modification requirements, we conclude that this article has the following innovations or contributions in the part of Disscusion:

This paper contributes to the current studies in four ways. (1) We use the survey data of 537 rural households in 50 villages in Yunnan Province, which is relatively underdeveloped in Southwest China and are located in Yunnan-Kweichow Plateau, mainly in plateau and mountain terrain, to study the impact of farmland transfer on rural household consumption, which has unique regional characteristics and greater practical significance. (2) By constructing an analytical framework of “farmland transfer - farmland function - income structure - rural household consumption”, we comprehensively analyzed the theoretical mechanism relationship between farmland transfer and rural household consumption. (3) Although there is a small amount of literature on the impact of farmland transfer on rural household consumption, this paper more systematically studies the impact of farmland transfer on rural household consumption through benchmark regression, robustness test and intermediary effect test. At the same time, we have achieved more fruitful study results. (4) Based on the empirical study of 537 rural households in 50 villages in Yunnan Province, we have obtained some new findings. For example, rural household consumption expenditure will show a downward trend with the increase of the age of the head of rural household, and the consumption structure will also show a deterioration. Another example is that the more family assets, rural households have the stronger consumption expenditure capacity, which is conducive to optimizing their consumption structure.

Finally, We would like to thank the third peer review expert for the constructive comments. We hope that our revised article can be published in the journal of Land very much.

Kind regards,

Mingyong Hong, Lei Lou.

Round 2

Reviewer 2 Report (New Reviewer)

thank you. This manuscript has been revised as requested.

This manuscript is a resubmission of an earlier submission. The following is a list of the peer review reports and author responses from that submission.

Round 1

Reviewer 1 Report

The present study researches the impact of farmland transfer on rural household consumption. The topic is important for the guidance of farmland transfer. However, some obvious mistakes are involved in the study.

1. In the Introduction section, what are the scientific questions on the impact of farmland transfer on rural household consumption? Why the impact of farmland transfer on rural household consumption should be studied? A general guess that there may be a certain correlation between the transfer of farmland and the consumption of rural households is not sufficient for the scientific question put forward.

2. From the review in lines 82-106, previous studies have analyzed the impact of farmland transfer on rural household consumption. What is the innovation of the present study?

3. The present study takes Yunnan Province, China as the study area. What is the typicalness? The Yunnan Province is not the major grain-producing area in China. What is the characteristic of farmland use and transfer in Yunnan Province?

4. For the impact of farmland transfer on rural household consumption, farmland transfer had heterogeneous effects on the consumption level of rural households with different characteristics such as the dependence on agricultural income but the details were ignored.

The current version of the paper is poorly organized and needs more work to satisfy readers.

Reviewer 2 Report

The presented paper is devoted to Research on the Impact of Agricultural Land Transfer on Rural Household Consumption: Evidence from Yunnan Province, China.

The individual parts are arranged logically, albeit in too much detail.

The authors of the article conduct a questionnaire survey. It is implemented on the basis of sample data of 537 rural households in 50 villages.

Important results presented:
• The transfer of agricultural land and its impact on stimulating rural households
• Coefficient of transfer of agricultural land to - non-food consumption - to food consumption
• As well as the results from the considered statistical model.

The individual parts of article are:
 1. Introduction; 2. Theoretical Analysis and Research Hypotheses; 3. Research Design; 4. Results and Analysis; 5. Robustness Test and Endogeneity Discussion; 6. Mechanism of Action: Intermediary Effect Test; 7. Conclusions and Policy Implications.

I have comments to the post:
Why so many chapters and subchapters. Couldn't you simplify the division of the article?
Line 116 From where the data in Figure 1 was considered. Write processed according to X (link to literary source).
Lines 118 to 144 - references should be sequential. The author skips numbers 1, 21, 9-12, 35-37, 38-39.
Line 216 It is not clear where do you enter to the scheme. If it is "farmland transfer", put an arrow there for entry. The result is "total consumption of rural households"? It is not clear.
Line 223 Surveys were conducted in 16 prefecture-level cities or autonomous prefectures in Yunnan Province. It was influenced by something, or it was the authors' idea. Can it be justified somehow?
Line 231 A total of 650 questionnaires were issued in this survey, and questionnaires with obvious errors, duplicate content and discrepancies with the research topic of this paper were invalid, resulting in a total of 537 valid questionnaires covering 50 municipalities with an effective rate of 82.62%. Why are you writing link 5) below the line. Can't you write it directly in the text? This is how it gets complicated.
It is similar
Line 238 reference to 6)
Line 241 reference to 7)
Line 248 reference to 8)
I state again: Can't you write it directly in the text? This is how it gets complicated.
Line 273 the statistical reasoning itself was done by the authors, it is difficult to check, it can be considered as it is OK.
Line 281 From line 3281, the author points to model selection. OK
Line 305 From 305 the results and their analysis are found. OK
Line 310 again a reference to 10). It is questionable how the authors meant it. Below is the following: Due to space limitations, the variation inflation factor (VIF) values ​​of the relevant variables are not shown; ask the author if necessary. This means that anyone who reads the article and is interested in this value should ask the author to send it to them?
These reminders are for all appeals. In my opinion, it would be better to put it directly in the text. See, for example, reference 14 - this is the full explanatory text: This means that the path of influence "agricultural land transfer - rural households" ... and the interpretation below the line. (Unnecessarily complicated). Similarly 15), 16).
Lines 279, 366, 383, 397, 421, 443, 500; There are tables there. Couldn't you present the results any way (f.e.graphical) as well?
The results presented in the tables could not be presented in graphs, for example? It would be clearer than numerical display of results in tables. The article has only two images. If the outputs were processed graphically, it would be more attractive for the reader.

I recommend publishing after necessary adjustments.